# Vasculonecrotic Reaction Caused by Mycobacterium Lepromatosis Infection—A Case Report of an HIV/Leprosy-Coinfected Patient

**DOI:** 10.3390/idr17030058

**Published:** 2025-05-23

**Authors:** Fernando Amador-Lara, Jorge L. Mayorga-Garibaldi, Felipe J. Bustos-Rodríguez, Luz A. González-Hernández, Pedro Martínez-Ayala, Jaime F. Andrade-Villanueva

**Affiliations:** 1Departamento de Clínicas Medicas, Centro Universitario de Ciencias de la Salud, Universidad de Guadalajara, Guadalajara 44280, Mexico; luceroga08@gmail.com (L.A.G.-H.); pemayala4@gmail.com (P.M.-A.); drjandradev@gmail.com (J.F.A.-V.); 2Unidad de VIH, Hospital Civil de Guadalajara Fray Antonio Alcalde, Guadalajara 44280, Mexico; 3Unidad de Diagnóstico en Microbiología Médica y Enfermedades Infecciosas, Guadalajara 44340, Mexico; leonardo_maygar91@hotmail.com; 4Departamento de Microbiología y Patología, Centro Universitario de Ciencias de la Salud, Universidad de Guadalajara, Guadalajara 44280, Mexico; fbustos@hcg.gob.mx

**Keywords:** vasculonecrotic reaction, leprosy, HIV infection, *Mycobacterium lepromatosis*, necrotizing erythema nodosum leprosum, Lucio’s phenomenon, leprosy reactions

## Abstract

Background: Vasculonecrotic reactions in leprosy are typically associated with type 2 reactions. Differentiating between necrotizing erythema nodosum leprosum (nENL) and Lucio’s phenomenon (LP) can be difficult, as overlapping clinical and histopathological features have been reported. *Mycobacterium lepromatosis*, a recently identified species causing leprosy, has been sporadically linked to LP. While type 1 reactions are more commonly observed in HIV-coinfected individuals, reports of LP or ENL occurring outside the context of immune reconstitution inflammatory syndrome (IRIS) remain rare. Methods: We report a case of a vasculonecrotic leprosy reaction due to *M. lepromatosis* in an antiretroviral-naive patient with advanced HIV infection. Results: The patient presented with a two-month history of papules and nodules that progressed to painful necrotic ulcers, accompanied by systemic symptoms. Clinically, the presentation was consistent with nENL; however, histopathological analysis supported a diagnosis of LP. The patient rapidly deteriorated, developing septic shock and dying shortly thereafter. To our knowledge, this is the first reported case of a leprosy-associated vasculonecrotic reaction caused by *M. lepromatosis* in an HIV-infected individual not associated with IRIS. Conclusions: Vasculonecrotic reactions in leprosy are life-threatening emergencies due to their potential for rapid clinical deterioration and sepsis. In individuals with advanced HIV infection, recognition of these reactions may be challenging, as they can mimic other opportunistic infections, including fungal diseases, malignant syphilis, and disseminated mycobacterial infections. Early identification and prompt treatment are critical to improving outcomes.

## 1. Introduction

Leprosy is a chronic granulomatous infection caused by *Mycobacterium leprae* or *Mycobacterium lepromatosis*. These two closely related species share approximately 87% nucleotide sequence similarity [1] and are believed to have diverged from a common ancestor around 13.9 million years ago [2].

Leprosy reactions are immune-mediated inflammatory episodes that may occur before, during, or after antileprosy treatment. These reactions can be triggered by *M. leprae* antigens, such as phenolic glycolipid-I, or by coinfection [3].

Leprosy reactions are broadly classified into two main types. Type 1 reaction (T1R), or reversal reaction, is a cell-mediated delayed-type hypersensitivity response. It typically manifests as the appearance of new skin lesions or the worsening of existing ones, often accompanied by painful swelling and thickening of peripheral nerves. T1R occurs predominantly in patients with tuberculoid, borderline tuberculoid, or borderline forms of leprosy. On the other hand, Type 2 reaction (T2R), or erythema nodosum leprosum (ENL), is characterized by the formation and deposition of circulating immune complexes in various tissues, most notably the skin, kidneys, and joints [3]. T2R is most commonly seen in patients with borderline lepromatous (BL) and lepromatous (LL) leprosy, both of which are immunologically unstable and prone to reactional episodes [4].

Vasculonecrotic reactions represent severe variants of T2R and include two key entities: a severe form of necrotizing erythema nodosum leprosum, which occurs in borderline lepromatous and lepromatous leprosy cases [5,6], and Lucio’s phenomenon, a leprosy reaction that occurs in an untreated anergic form of the disease known as diffuse lepromatous leprosy, although it can also present in borderline lepromatous and lepromatous leprosy forms [5]. Differentiating between nENL and LP can be clinically and histopathologically challenging [7,8].

We report a fatal case of a vasculonecrotic reaction in an HIV/leprosy-coinfected patient, caused by *Mycobacterium lepromatosis*.

## 2. Case Presentation

A 33-year-old male from Jalisco, residing in Aguascalientes, Mexico, with a recent diagnosis of HIV infection, presented with a two-month history of violaceous papules and nodules that progressively evolved into painful necrotic ulcers. The lesions initially appeared on the trunk and later spread to the extremities and face. Systemic symptoms included malaise, unintentional weight loss, fever, and nocturnal diaphoresis. Fifteen days prior to admission, the patient experienced sudden vision loss in the right eye. Physical examination revealed multiple deep, punched-out ulcers ranging from 5 to 20 mm in diameter, round to ovoid in shape, with regular borders (Figure 1). Ophthalmologic evaluation confirmed complete vision loss in the right eye, with retinal detachment and hemorrhagic and fibrotic lesions compatible with cytomegalovirus (CMV) chorioretinitis. Intravenous ganciclovir therapy was initiated accordingly.

Laboratory tests showed the following abnormalities: hemoglobin 9.4 g/dL, hematocrit 27.9%, leukocytes 3.76 × 10^3^/µL, neutrophils 3.16 × 10^3^/µL, lymphocytes 0.21 × 10^3^/µL, monocytes 0.33 × 10^3^/µL, eosinophils 0.20 × 10^3^/µL, basophils 0.06 × 10^3^/µL, and platelets 187 × 10^3^/µL. Additional findings included elevated liver enzymes (ALT 199 U/L, AST 331 IU/L, GGT 176 IU/L, alkaline phosphatase 215 IU/L), hypoalbuminemia (2.4 g/dL), elevated lactate dehydrogenase (486 U/L), ferritin (475 ng/mL), C-reactive protein (12.6 mg/dL), and procalcitonin (0.25 ng/mL). VDRL was reactive with a titer of 1:2. Abdominal ultrasound showed hepatomegaly with fatty infiltration; no other significant findings were reported. Serologic testing was negative for toxoplasmosis, hepatitis B and C viruses, and cryptococcal antigenemia. CMV IgG was positive; however, CMV DNA quantification was not performed. Blood cultures were negative for bacteria, fungi, and mycobacteria. A contrast-enhanced chest, abdominal, and pelvic CT scan revealed no relevant abnormalities. Initial HIV-1 RNA viral load was 301,000 copies/µL, and CD4+ T-cell count was 4 cells/mm^3^. Antiretroviral therapy with bictegravir, emtricitabine, and tenofovir alafenamide was initiated during hospitalization.

A skin biopsy of a representative lesion showed epidermal ulceration and necrosis with a prominent acute inflammatory infiltrate composed predominantly of polymorphonuclear cells. The dermis exhibited a dense, chronic histiocytic infiltrate with neurotropic distribution. Additionally, the subcutaneous tissue demonstrated a lobular panniculitis pattern, leukocytoclastic vasculitis, and vascular thrombosis. Immunohistochemical staining for CD68 confirmed the presence of perineural macrophages. No viral cytopathic changes were identified. Fite–Faraco staining revealed abundant acid-fast bacilli. Based on the histopathological features, a diagnosis of lepromatous leprosy with Lucio’s phenomenon was established (Figure 2). Additional tests performed on the biopsy, including Grocott methenamine silver staining, fungal and mycobacterial cultures, and Xpert MTB/RIF, were all negative.

A PCR assay was performed on the skin biopsy tissue to identify the causative microorganism. Amplification was carried out using 1% agarose gel electrophoresis and the LPMF-244 primer set: LPM244-F [5′–3′ GTTCCTCCACCGACAAACAC] and LPM244-R [5′–3′ TTCGTGAGGTACCGGTGAAA], targeting the *hemN* gene region—specific to *Mycobacterium lepromatosis* and absent in other known *Mycobacterium* species (Figure 3).

On the fifth day of hospitalization, the patient developed septic shock, which was managed with broad-spectrum antibiotics (meropenem and vancomycin) and vasopressor support. Despite these measures, he showed no clinical improvement and ultimately succumbed to septic shock. An autopsy was not authorized by the family. Importantly, antileprosy treatment was not administered, as the definitive diagnosis was confirmed one day after the patient’s death.

## 3. Discussion

This report describes a case of a vasculonecrotic reaction with systemic manifestations due to *Mycobacterium lepromatosis* infection in an HIV-infected individual with severe immunosuppression, who was unaware of having leprosy and naïve to antiretroviral therapy. The patient experienced rapid clinical deterioration and ultimately succumbed to sepsis.

Vasculonecrotic lesions in leprosy occur as part of type 2 leprosy reactions, most frequently as a complication of the anergic form of diffuse lepromatous leprosy [8,9]. Differentiating between necrotizing erythema nodosum leprosum and Lucio’s phenomenon can be challenging, with reported cases of overlapping features [5,7,8,10]. Clinical and histopathological findings are key to establishing the diagnosis [11].

Necrotizing erythema nodosum leprosum typically presents as nodular lesions that evolve into painful necrotic ulcers and is associated with systemic symptoms such as fever, myalgia, arthralgia, malaise, lymphadenopathy, iritis, episcleritis, hepatitis, neuritis, and orchitis. It generally occurs after the initiation of anti-leprosy treatment [7]. In contrast, Lucio’s phenomenon is characterized by multiple, extensive purpuric lesions with a reticular distribution and superficial ulceration over infiltrated skin [5]. These polygonal ulcers, with angulated and jagged borders, often progress to necrosis, predominantly affecting the extremities. The lesions are associated with a burning sensation and evolve into stellate, atrophic scars. Systemic symptoms and visceral involvement are typically absent; however, extensive skin damage may predispose patients to secondary bacterial infections and sepsis, leading to symptoms related to these additional infectious complications. Lucio’s phenomenon occurs in untreated or inadequately treated cases of non-nodular lepromatous leprosy [7] and often represents the initial manifestation of the disease [11]. Our patient did not have a previous diagnosis of leprosy. In a systematic review, 85.7% (42/49) of Lucio’s phenomenon cases had no prior diagnosis of leprosy [12].

Histologically, necrotizing erythema nodosum leprosum is characterized by dermal histiocytic infiltrates, deep dermal neutrophilic infiltration, vasculitis, and panniculitis with sparse bacillary presence [12]. Lucio’s phenomenon, on the other hand, demonstrates abundant acid-fast bacilli within vascular endothelium, leukocytoclastic vasculitis, endothelial proliferation [8], ischemic epidermal necrosis [12], necrotizing vasculitis of superficial and medium-sized vessels [13], and fibrin thrombi [7].

The pathogenesis of LP involves massive bacillary infiltration of vascular endothelium, leading to luminal narrowing, thrombosis, and cutaneous necrosis. This is followed by an inflammatory response and immune complex deposition [10]. Immunofluorescence studies have shown deposits of IgM, IgG, C3, and C1q in dermal blood vessels [14]. Differential diagnosis includes cryoglobulinemia, leukocytoclastic vasculitis, pyoderma gangrenosum, antiphospholipid syndrome, and disseminated intravascular coagulation [15,16]. Erythema nodosum leprosum, in contrast, is triggered by immune complex deposition [10].

Due to the high bacillary load in LP, prompt initiation of multidrug therapy is essential, often in combination with corticosteroids, antibiotics, anticoagulants, surgical debridement, and skin grafting [17,18]. Necrotizing ENL, however, typically responds to corticosteroids and thalidomide [5].

*M. lepromatosis* was identified in 2008 as a distinct species causing fatal diffuse lepromatous leprosy in two Mexican patients [19]. While endemic in Mexico, *M. lepromatosis* has also been reported in other parts of the Americas and sporadically in Asia [20,21,22]. Its clinical presentation most often involves diffuse lepromatous leprosy, an endemic Mexican form characterized by diffuse, non-nodular infiltration, originally described by Lucio and Alvarado in 1852 [23] and later revisited by Latapí and Chevez-Zamora in 1948 [24].

In a study by Han et al., among 87 species-confirmed leprosy cases in Mexico, *M. lepromatosis* was identified in 63.2%, *M. leprae* in 20.7%, and both species in 16.1% of cases. Among those with *M. lepromatosis*, 61.8% had lepromatous leprosy, and 23.6% presented with diffuse lepromatous leprosy [25]. These findings differ from a Brazilian study, in which all seven *M. lepromatosis*-positive patients had tuberculoid leprosy, suggesting a possible role of host genetics or ethnicity in clinical expression [26].

Both *M. leprae* and *M. lepromatosis* have been identified by PCR in cases of LP, indicating that host factors may play a more crucial role in the development of this reaction than the specific species involved [27]. Most reported LP cases associated with *M. lepromatosis* involve patients from Mexico or the Caribbean [12,28,29,30].

HIV infection has dramatically changed the epidemiology of several mycobacterial diseases, notably *Mycobacterium tuberculosis* and *Mycobacterium avium* complex, both of which are more prevalent and severe in HIV-infected individuals [31]. However, HIV does not appear to increase leprosy incidence, and no significant difference in HIV seroprevalence has been found between newly diagnosed leprosy cases and controls [32].

Given the impairment of cell-mediated immunity in HIV, an increased frequency of lepromatous leprosy in coinfected individuals was anticipated. However, the clinical spectrum of leprosy remains unchanged in HIV-infected individuals [33], and histopathological features are not significantly altered. A comparative study found no differences in histological features between coinfected and mono-infected individuals, with typical granulomatous responses still observed in HIV-positive patients [34]. Granuloma formation remains preserved even in patients with low CD4+ T-cell counts [35].

An increased incidence of leprosy has been observed during the first three months following initiation of antiretroviral therapy (ART), likely due to immune reconstitution. This risk normalized after three months of antiretroviral therapy [36]. Several cases of paradoxical immune reconstitution inflammatory syndrome (IRIS) have been reported, predominantly presenting as T1R within six months of ART initiation, especially in borderline tuberculoid leprosy [37,38], principally manifesting as leprosy T1R after improved cell-mediated immunity due to increasing the CD4+ T lymphocyte count, usually within the first six months of antiretroviral therapy, mostly in borderline forms, predominantly the borderline tuberculoid form [39].

Although increased rates of leprosy reactions have been observed in HIV/leprosy-coinfected individuals [40,41], a cohort study comparing 40 coinfected patients with 107 HIV-negative leprosy patients found no significant difference in the frequency of leprosy reactions. In the coinfected group, 86.7% of reactions were T1R, 53.3% occurred as part of IRIS, and 93.3% were in the AIDS stage, suggesting multifactorial influences on immune behavior. No significant clinical or histopathological differences were noted between coinfected and non-infected groups [42].

In the largest cohort of coinfected individuals in Brazil, 33 of 92 patients (36%) experienced leprosy reactions at diagnosis, 97% of which were T1R. ART use was the only factor associated with T1R [43]. Although T1R is more common, isolated cases of LP and ENL have been reported in the context of unmasking or paradoxical IRIS [44,45,46].

Lucio’s phenomenon is a life-threatening medical emergency [47], with sepsis being a frequent and often fatal complication due to secondary bacterial infection of necrotic skin lesions [48]. In a systematic review, 30.6% (15/49) of patients with LP received systemic antibiotics for sepsis, with high mortality reported [12]. In our case, the patient died of sepsis. Although no pathogen was isolated, we believe the sepsis was likely bacterial, entering through the extensive skin lesions.

Our case presented with nodular lesions that progressed to deep, painful necrotic ulcers accompanied by systemic symptoms—clinically resembling necrotizing erythema nodosum leprosum. However, histopathological features were consistent with Lucio’s phenomenon, including epidermal necrosis, vasculitis, thrombosis, and abundant acid-fast bacilli in a patient with untreated diffuse lepromatous leprosy. We postulate that the profound deficiency in cell-mediated immunity due to HIV coinfection and an extremely low CD4+ T-cell count allowed for an overwhelming bacillary burden, ultimately triggering the pathophysiological mechanisms characteristic of Lucio’s phenomenon.

## 4. Conclusions

Vasculonecrotic reactions in leprosy represent a medical emergency associated with significant morbidity and mortality. To our knowledge, this is the first reported case of a vasculonecrotic reaction due to *Mycobacterium lepromatosis* in an HIV-infected subject, not associated with immune reconstitution inflammatory syndrome. In HIV-infected individuals, leprosy reactions may be challenging to recognize, particularly in the setting of severe immunosuppression, where atypical and severe presentations of other opportunistic infections—such as cutaneous cryptococcosis, disseminated sporotrichosis, malignant syphilis, or cutaneous mycobacterial infections—can mimic or obscure the clinical picture. Delayed diagnosis and initiation of appropriate therapy can result in extensive cutaneous damage, secondary bacterial infection, sepsis, and increased risk of death.

## Figures and Tables

**Figure 1 idr-17-00058-f001:**
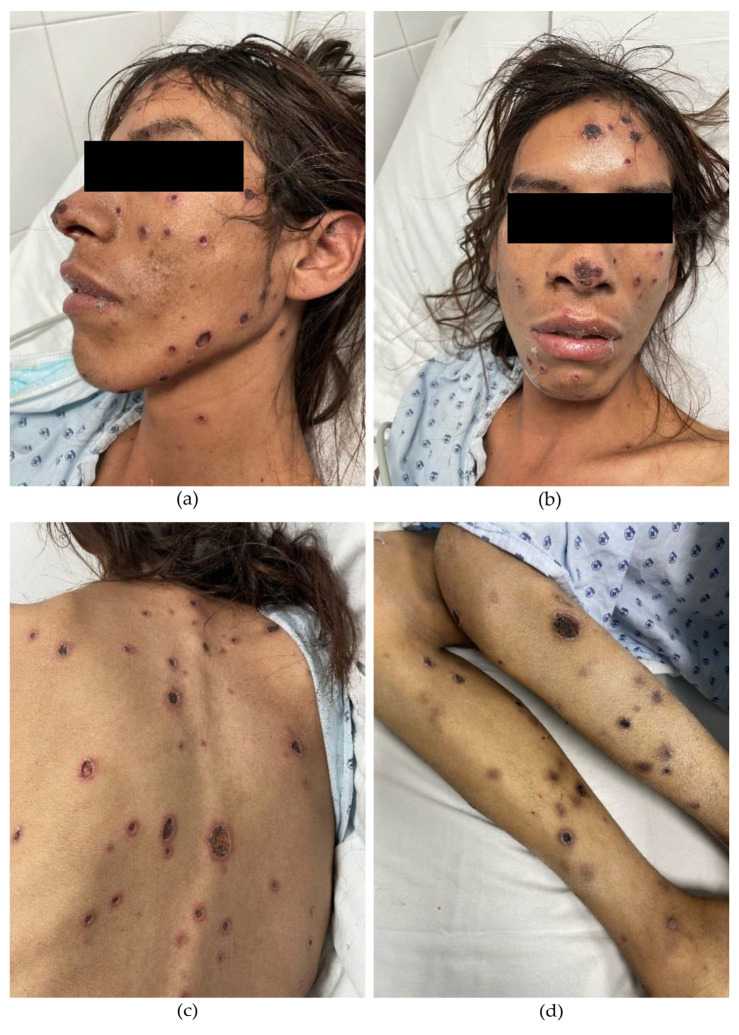
(**a**,**b**) Multiple deep, well-demarcated, punched-out ulcers covered with necrotic eschars. Several papules and nodules are observed in transition to necrotic ulcers on the face. (**c**) Ulcers on the back display an erythematous halo. (**d**) Ulcers on the extremities appear in a later stage of evolution.

**Figure 2 idr-17-00058-f002:**
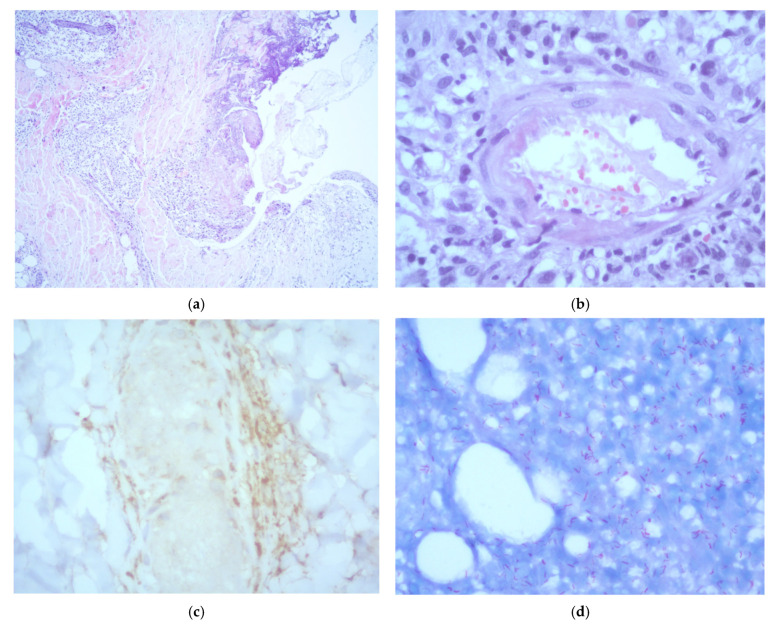
Histopathological examination of a skin biopsy showing: (**a**) epidermal reactive changes with ulceration and necrosis, accompanied by an acute inflammatory infiltrate composed predominantly of polymorphonuclear neutrophils (H&E, ×5); (**b**) evidence of leukocytoclastic vasculitis (H&E, ×40); (**c**) CD68 immunohistochemistry showing positive staining of perineurally located macrophages (×40); (**d**) numerous acid-fast bacilli observed both within and outside macrophages (Fite–Faraco stain, ×40).

**Figure 3 idr-17-00058-f003:**
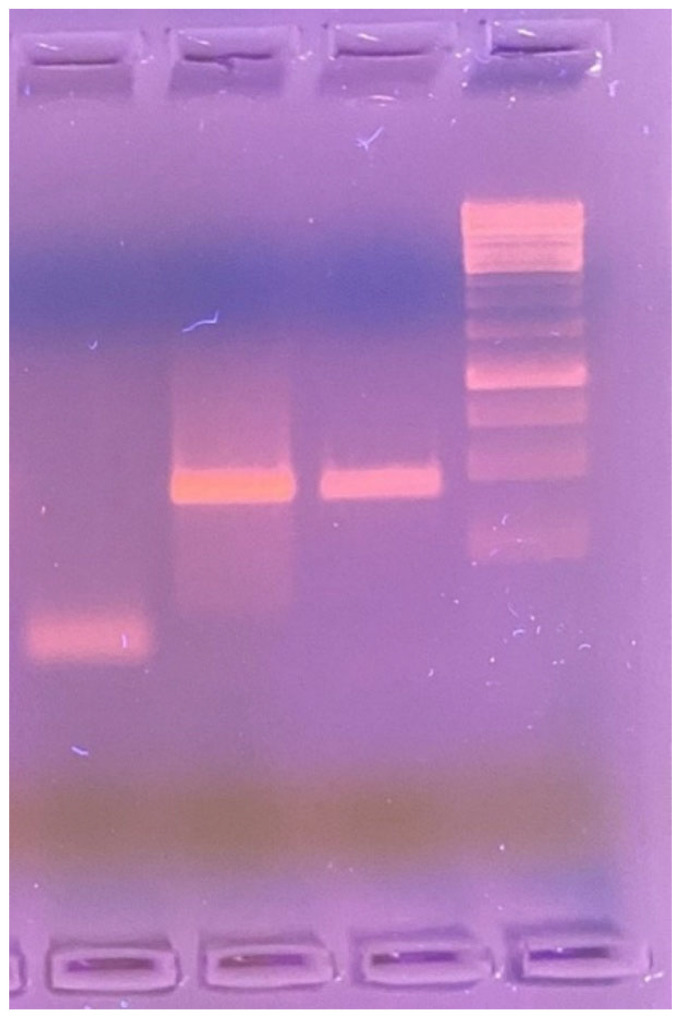
1% agarose gel electrophoresis (left to right): negative control—sterile saline solution; positive control—DNA sample from a patient with confirmed *Mycobacterium lepromatosis* infection; test sample—PCR product amplified using LPMF-244 primers targeting a *hemN* gene region specific to *M. lepromatosis* (100 bp), absent in other known *Mycobacterium* species.

## Data Availability

All data are included within the manuscript.

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
