# Peer review of "Vasculonecrotic Reaction Caused by Mycobacterium Lepromatosis Infection—A Case Report of an HIV/Leprosy-Coinfected Patient"

_2036-7449, 2025, doi:10.3390/idr17030058_

Round 1
Reviewer 1 Report
Comments and Suggestions for Authors
The manuscript is well written and presents an interesting and complex case. I have several comments and questions for the authors' consideration:
The patient described is HIV-positive and presented with clinical features suggestive of either a type 2 lepra reaction (T2R) or Lucio’s phenomenon. As the authors correctly note, Lucio’s phenomenon is typically not associated with systemic symptoms. However, in this case, the patient exhibited significant systemic involvement. Were alternative causes of systemic manifestations—such as disseminated cytomegalovirus (CMV) infection or toxoplasmosis—adequately excluded? For example, was CMV DNA tested in blood or were blood cultures performed?
The patient reportedly died of sepsis. Could the authors clarify whether the sepsis was considered to be a direct consequence of leprosy? Providing more detailed laboratory data from admission—such as procalcitonin, C-reactive protein (CRP), ferritin levels, and complete blood count (CBC)—would help to better characterize the systemic inflammatory response.
Were any imaging studies, such as computed tomography (CT) or magnetic resonance imaging (MRI), performed as part of the diagnostic workup? Furthermore, were molecular analyses conducted on the skin biopsy to assess for potential co-infections?
It would also be valuable to include more detailed information on the patient’s past medical history, the length of hospital stay prior to death, and whether blood cultures were performed and what results, if any, were obtained. Did the patient undergo an autopsy? Finally, it remains unclear whether the patient received any specific anti-leprosy treatment. Clarification on this point would contribute significantly to the completeness of the clinical narrative.
Comments on the Quality of English LanguageIt seems fine to me, anyway I am not in a position to assess the quality of the English language used in the manuscript, as I am not a native speaker.
Reviewer 2 Report
Comments and Suggestions for Authors
Manuscript Vasculonecrotic reaction in an HIV/leprosy coinfected subject caused by Mycobacterium lepromatosis infection by Amador-Lara et al. represents a case report article describing coinfected patient with these infective agents and corresponding clinical events and outcomes. In general, the paper is written nicely. I have several suggestions for Authors and they are as following:
- The order of the words in title is somewhat strange-I assume that this is due to English not being native to Authors, and the different rules of Spanish. I suggest to Authors the following title: Vasculonecrotic reaction caused by Mycobacterium lepromatosis infection - a case report of HIV/leprosy coinfected patient. It is always good to point out that article is a case report, I suggest patient instead of subject as you did treat him in clinics, and this order of words seems better to me.
- Regarding the abstract- although I find all said and written in Conclusion part to be fine and true, please make it a bit shorter. Section is disproportional to other part, and if any should be bigger and longer than others, it is results, rather than conclusions.
- In Case presentation segment- please add whether did patient had antiretroviral drugs, and if yes, add the regimen. Any other types of drugs? What was the reason for patient's hospitalization? line 88- I believe you have a typo in title 1:1? Also, US should be defined.
- Discussion- Line 133 answers my question about cART "naive to antiretroviral treatment,"--why was this the case? The general recommendation for all discovered cases is to start treatment as soon as possible. Otherwise, this segment is excellently written- I had several ideas what would I like to ask, and all the answers were given in Discussion segment. I would only asked for info- how frequent is leprosy in Mexico? Is it a frequent or rare disease?
- Conclusion is fine, no remarks there.
- Informed Consent Statement: Informed consent for publication was obtained from the patient for the case report and imaging.--- Did you actually obtained permission from the patient or the consent was obtained from a family member? I am interested as I had not had deceased patients in my case reports so I do not know these regulations?
Kind regards, and best of wishes in publishing your paper!
Reviewer 3 Report
Comments and Suggestions for Authors
The authors present a novel case of patient with HIV/AIDS and co infection with m. lepromatosis with a clinical presentation regarded and treated as Lucio phenomena.
General comment for the introduction:
There is a lot of usage of abbreviations, that are not used afterwards in the text and it makes difficult to follow and read. Maybe reconsider to keep some of them but not all of them. I miss good information on the pathophysiological mechanism between ENL and LP. This is important to understand the evolution of the disease, I think.
Case description:
- Did the patient start ART? Why was it delayed/not started? How do you justify?
- Did you start treatment for lepromatosis?
Specific comments:
Line 116 and 124 -> Typing mistake-> primers LPM244 F and R? or in general LPM 244?
Line 117 -> HemN gene is also present in M tuberculosis. Was in this patient a co infections suspected? If not why not?
Line 170- there is no information on the approach taken with this patient. Which treatment was started besides vancomycine and meropenem? The patient had AIDS, were there no other possible co infections? Crypto/ TB/ HBV/ PJP/ other atypical infections?
Line 212: ‘’…an increased incidence of leprosy ..? was observed …’’ leprosy skin leasons, reactions?.. please clarify
Lines 148 vs info in line 235 – contradictory. Line 148 says …’’ systemic symptoms and visceral involvement are usually absent..’’ and line 235 says ..’’ life threatening condition’’.. please clarify, best in lines 148.
Line 238: Please clarify the information on: ..’’patient died of sepsis’’. Were there pos haemocultures? Was a case of disseminated m. lepromatosis with invasion in blood of elsewhere, were other co infections etc?.
